# Swimmers with Down Syndrome Are Healthier and Physically Fit than Their Untrained Peers

**DOI:** 10.3390/healthcare11040482

**Published:** 2023-02-07

**Authors:** Ana Querido, Mário J. Costa, Dália Araújo, António R. Sampaio, João P. Vilas-Boas, Rui Corredeira, Daniel J. Daly, Ricardo J. Fernandes

**Affiliations:** 1N2i, Polytechnic Institute of Maia, 4475-690 Maia, Portugal; 2Centre of Research, Education, Innovation and Intervention in Sport, CIFI2D, Faculty of Sport, University of Porto, 4200-450 Porto, Portugal; 3Porto Biomechanics Laboratory, LABIOMEP-UP, Faculty of Sport, University of Porto, 4200-450 Porto, Portugal; 4Research Centre in Physical Activity, Health and Leisure, CIAFEL, Faculty of Sport, University of Porto, 4200-450 Porto, Portugal; 5Faculty of Movement and Rehabilitation Sciences, Katholiek Universiteit Leuven, 3001 Leuven, Belgium

**Keywords:** physical fitness, body composition, health, measurement, Down syndrome

## Abstract

While there are positive benefits from physical activity participation for individuals with Down syndrome, little is known about the effects of swimming training. The aim of this study was to compare the body composition and physical fitness profile of competitive swimmers and moderately active (untrained) individuals with Down syndrome. The Eurofit Special test was applied to a group of competitive swimmers (*n* = 18) and a group of untrained individuals (*n* = 19), all with Down syndrome. In addition, measurements were taken to determine body composition characteristics. The results showed differences between swimmers and untrained subjects in height, sum of the four skinfolds, body fat %, fat mass index and all items of the Eurofit Special test. Swimmers with Down syndrome exhibited physical fitness levels near to the Eurofit standards, although lower fitness levels were attained by these persons when compared to athletes with intellectual disability. It can be concluded that the practice of competitive swimming seems to counteract the tendency for obesity in persons with Down syndrome and also helps to increase strength, speed and balance.

## 1. Introduction

Down syndrome, a form of intellectual disability, is a genetic disorder caused by the presence of the whole (or part) of an extra copy of chromosome 21, with a global incidence estimation of 1 in 1000 to 1 in 1200 live births [1]. These individuals, present distinctive physical features, are predisposed to a higher incidence of cardiovascular disease [2], diabetes [3], osteoporosis and obesity, and more susceptible to a premature and significant decline in function with age [4]. Despite this, the infant Down syndrome survival rate, as well as life expectancy in general, continues to increase [5,6].

Lack of regular physical activity has been identified as one of the most significant health risks and people living with chronic conditions or disability are now being given recommendations for the first time [7]. This inactivity results in an increased threat of chronic conditions (e.g., cardiovascular disease and type 2 diabetes) [8] and is considered as a predictor of mortality in the Down syndrome population [9]. Literature indicates low fitness levels in these individuals [10,11,12], which may be related to sedentary lifestyles [13], limited social and recreational opportunities [14] and/or low motivation to be physically active [15]. Nevertheless, several studies have indicated positive benefits from physical activity participation for these individuals [16,17,18] and evidence suggests that physical activity can increase their physical fitness [19]. Specifically, aquatic exercise has been shown to offer benefits for people with intellectual disabilities in terms of cardiorespiratory endurance, muscular endurance, speed, static balance and agility [20,21].

There has been an increasing interest from people with Down syndrome in competitive swimming, with the participation of ~200 swimmers at the 2022 World Championships held in Portugal. Even if physical activity and sport are meaningful to many people, including those with intellectual disabilities, research in this topic has focused mainly on inactive participants [22], while trained individuals are scarcely studied [23]. In the specific case of competitive swimming, one study showed that an 18 week training intervention did not promote clear changes in jumping performance or body composition in swimmers with Down syndrome [24]. In a contrasting study, 33 weeks of swimming training lead to improved health status and swimming skills [25]. So, some mixed findings exist on how swimmers with Down syndrome may react to swimming training and exhibit improved physical form in comparison to their non-swimmer peers. This lack of evidence makes us question if swimmers with Down syndrome, even those at the top level, still remain healthier, or show better physical condition than their peers who are not involved in any intensive sport participation.

Thus, the aim of this study was to assess the body composition and the physical fitness profile of competitive swimmers with Down syndrome and to compare them to untrained peers. It was hypothesized that: (i) swimmers would present lower values than untrained individuals for Body Mass Index (BMI), percentage of total body fat (fat%) and Fat Mass Index (FMI), and higher values for Lean Body Mass (LBM); and (ii) swimmers would also present higher physical fitness values than untrained counterparts.

## 2. Materials and Methods

### 2.1. Participants

To be elegible for this study, trained participants had to be, as minimum, national competitive swimmers, being involved in a minimum of 6 h of swimming training per week, for at least 3 years. To be a part of the control group, participants could not be involved in any kind of competitive sport practice for the last 3 years.

Thirty-three individuals with Down syndrome participated in this study: 18 were national or international level trained swimmers and 19 were untrained persons with Down syndrome. Trained swimmers were 22.2 ± 5.4 years, practiced 7.4 ± 0.8 h per week over the entire year and had been involved in swimming training for several years. More than half of the participants were part of the National Federation Swimming Team and participated in DSISO International Championships. Untrained individuals were 26.6 ± 8.2 years and were involved in two 45 min sessions a week of general physical activity. All individuals, or their parents, gave written informed consent to participate in this study, which was approved by the local ethics committee (under the code: CEFAD 19.2020) and carried out according to the Declaration of Helsinki. 

### 2.2. Procedures and Measures

The body measurements included height, weight and four skinfolds (triceps, biceps, subscapular and supra-iliac, using a Harpender skinfold caliper). BMI (in kg/m^2^) was defined as body mass (in kg, measured using an electronic weighing scale) divided by height squared (in m). The fat% and LBM were derived from the measured skinfolds, using the equation proposed by Durnin & Wommersley [26] and the FMI was calculated as fat mass/height^2^. All measurements were made on the right side of the body by the same evaluator and were repeated three times, with the mean values being used [27]. The classification of obesity was made according to the World Health Organization technical report as follows [28]: underweight if BMI ≤ 18.4, normal weight if BMI 18.5–24.9, overweight if BMI 25–29.9, obesity if BMI 30–39.9 and morbid obesity if BMI ≥ 40.

To evaluate physical fitness, the Eurofit Special test was used, as follows [29]: (i) explosive lower limb strength was determined with a standing broad jump; (ii) upper limb strength was determined using a 2 kg medicine ball push performed with the preferred arm—from a standing position, the ball was placed in the palm, supported by the second hand and pushed forward in a shot put like action; (iii) local muscle endurance was determined by the number of correctly completed sit-ups in 30 s; (iv) speed was measured for a 25 m run from a standing start measured to 0.1 s using a manual stopwatch; (v) flexibility was measured with the sit-and-reach test; and (vi) balance was determined by walking on a bench. Two test trials (Test A and Test B) were performed without shoes. In test A, the participant approached the bench, stepped onto it and walked forward. If Test A was successful, Test B is attempted. For test B the same process applies, with the bench in the upside-down position. Each test had to be completed in 30 s, with points recorded on the following scale: 1 point if the participant responds to the instructions; 2 points if the participant approaches the bench; 3 points if the participant walks 2 m without support or the entire bench with support (Test A); 4 points if the participant walks along the entire bench without support (Test A); 5 points if the participant walks 2 m without support or the entire bench with support (Test B); 6 points if the participant walks along the entire bench without support (Test B). A familiarization for all tests was allowed two weeks before the data collection to guarantee that the physical fitness tests were fully understood by the subjects and could therefore be carried out properly. 

The results from the body mass, height and all the Eurofit items were then converted to percentile scores. This facilitated merging of gender groups for this study (14 males and four females swimmers and nine males and 10 females untrained). The norm scales for severe intellectual disabled individuals of 20 years old (top off age for the scale) were used [30]. This table was chosen for two main reasons: (i) most raw scores measured fit this table, particularly the control group, and (ii) 20 years was the closest age to the sample studied. Scores outside the scale were given the maximum or minimum points, as appropriate.

### 2.3. Statistical Analysis

Descriptive statistics were calculated for all the variables (raw values as well as percentile scores) and all data were checked for normality and homogeneity of variance using Shapiro-wilk and Levene tests, respectively. Mean and SD for all variables are presented. An independent sample t-test was used to verify if there were differences between groups on performance and body composition (independently of the sex). Cohen’s d was calculated for effect size and interpreted as: small if d ≤ 0.2, moderate if d between 0.2 and 0.5, and large if d ≥ 0.5). The statistical significance was set at *p* ≤ 0.05. Procedures were performed with SPSS Statistics (v. 27, IBM, SPSS Inc., Chicago, IL, USA).

## 3. Results

Table 1 presents the comparison of body composition and all variables of the Eurofit Special Test between trained swimmers and untrained subjects. The swimmers presented higher values for height and lower values for the sum of the skinfolds, BMI, fat% and FMI, all with large effect sizes. Concerning sex differences, male swimmers and untrained individuals were taller and had higher values for LBM and lower values for fat% than female counterparts. In the swimmers group, 10 persons were of normal weight, six were overweight, and two were obese whereas in the untrained group, seven were of normal weight, four were overweight, six were obese, and two were morbid obese. Thus, 44.4 and 52.6% of the participants were overweight or obese in swimmers and untrained individuals, respectively and 10.5% of the untrained group showed morbid obesity. 

Swimmers with Down syndrome also scored better with large effect for long jump, medicine ball, sit-ups, speed and balance, while for flexibility there was a moderate effect. 

Table 2 presents the percentile scores that are based on norms for severe intellectual disability without Down syndrome. Swimmers presented higher scores for all variables (groups were not different in body mass), with large effect sizes, except for the medicine ball throw with a moderate effect. Despite those differences, both swimmers and untrained subjects showed low percentile scores for height (28.8 ± 14.5 for swimmers), and medicine ball (36.7 ± 25.3 for swimmers) on the scale used.

## 4. Discussion

The purpose of this study was to assess the body composition and the physical fitness profile of competitive swimmers with Down syndrome and to compare them with untrained individuals with the same condition. It was found that swimmers with Down syndrome present a healthier body composition and a higher physical fitness score than untrained individuals with Down syndrome.

### 4.1. Body Composition

Epidemiological studies often use BMI as a measure of weight status [31] since it is a good indicator of body fatness and easily calculated [32]. Nevertheless, several authors reported that BMI is not sufficient to describe the body composition of individuals [33,34,35]. Taking this into consideration, the fat mass index was also calculated here, due to the fact that fat mass is in part related to height [36]. Pitetti et al. [37] pointed out that ambiguous evidence exists regarding body composition in persons with Down syndrome and that the lack of consistency may involve methodological issues in measuring body composition or in the comparison of the weight status using different methods. Therefore, caution is urged when interpreting global statements on body composition in Down syndrome.

Numerous studies have reported that the prevalence of overweight and obesity are substantially higher in individuals with Down syndrome compared to their age-matched peers without disability, as well as those with intellectual disability but not related to Down syndrome [37]. Prasher [38] reported that ~48% of adults with Down syndrome were obese with ~27% being overweight and Rubin et al. [39] found ~48% of men and ~56% of women to be overweight or obese. In the present study, according to the BMI criteria, 55.6% of the swimmers and 36.8% of the untrained were normal weight, 33.3% of the swimmers and 21.1% of the untrained were overweight, 11.1% of the swimmers and 31.6% of the untrained were obese and 10.5% of the untrained were on the morbid obesity range. According to the fat% criteria, a much larger percentage of swimmers (83.3%) and a smaller percentage of untrained (31.6%) were considered normal. A much smaller percentage of swimmers were considered overweight (11.1%), while for the untrained the percentage rose to 63.2%, while 5.6% of the swimmers and 5.2% of the untrained were on the borderline range.

Although the equation described by Durnin & Womersley [26] to estimate body fat is not specific for Down syndrome, it was nevertheless previously used with this population [40]. These authors calculated the fat% in male adolescents with Down syndrome before and after a 12-week moderate aerobic training program. These adolescents decreased their fat mass percentage after the program (31.8 ± 3.7% pretest and 26.0 ± 2.3% posttest). We should note that male swimmers from our study presented much lower fat% than these adolescents (18.8 ± 3.7%).

In the current study, swimmers presented higher values for height and lower values for the sum of skinfolds, BMI, fat% and fat mass index than untrained peers. There was a large effect for all of the variables above, indicating that, not only swimmers are different from untrained, but that those differences are great. Gonzalez-Aguero et al. [19] stated that body composition in this specific population is, in general, poorer than what is observed in their peers without Down syndrome, as proven by higher BMI, lower levels of lean mass and reduced bone mass-related parameters. Bertapelli et al. [41] reported several causes for the augmented obesity in persons with Down syndrome, such as genetic, physiological, and environmental factors. However, as mentioned previously, the common term “obesity” used to describe physical characteristics in individuals with this condition might not always be valid [42]. For instance, a review by Gonzalez-Aguero et al. [19] reported mixed results and, if some studies indicate higher fatness values for people with Down syndrome [41,43,44,45,46], others present similar levels for persons with Down syndrome relative to persons without [42,43,47]. Despite these uncertainties, people with Down syndrome seem capable of improving their body composition values with training [40,44].

In athletes with Down syndrome or intellectual disability, systematic training seems to lead to healthier body composition, and, consequently, a better quality of life [19]. In a study from Aleixo et al. [48], with a small number of individuals with Down syndrome, differences in BMI between swimmers (24.3 ± 4.1) and untrained (36.8 ± 5.3) were observed, with swimmers being placed on the normal weight range and the untrained individuals at the obesity level. On the other hand, Balic et al. [16] analyzed 13 trained individuals who participated at the Special Olympics Games and seven sedentary adults, all with Down syndrome, and found no differences between both groups in BMI, height, weight and fat%. We may argue that, at the time of the study of Balic et al. [16] (approximately 20 years ago), the training for the Special Olympics started to be more demanding and not such an oriented and occupational practice. This training effect may also be the deciding factor for the differences between swimmers from our study and Climstein et al. [49]. These authors evaluated one group of 15 individuals with Down syndrome and one group of 17 non-Down syndrome and most of the subjects were actively involved in the Special Olympics program. When compared to the present study, Down syndrome individuals from Climstein et al. [49] presented higher values for fat% (26.1 ± 5.3%).

### 4.2. Physical Fitness

Swimmers presented better results for all the test items, indicating their better physical fitness profile. This higher score was confirmed when examining the percentile scores (Table 2) as swimmers with Down syndrome are more fit than untrained counterparts, presenting higher levels of strength, balance and flexibility.

Despite the fact that physical fitness is an important contributor to health in adults and youth, less is known in persons with disabilities, such as Down syndrome [42]. In a review on physical fitness and physical activity in children and adolescents with Down syndrome, Pitetti et al. [37] point out that peak aerobic capacity (VO_2peak_) in both youth and adults with Down syndrome is reduced in comparison to their peers without disability and with other intellectual disabilities. The same authors highlight the fact that these persons can be responsive to aerobic endurance training, particularly with improvements in work capacity. Muscular strength is also lower in persons with Down syndrome when compared to their peers with normal development or with other intellectual disabilities [11,17,37,50,51,52,53]. According to Shields et al. [54], an improved strength in persons with Down syndrome has been associated with higher levels of physical activity. Muscular strength is a fundamental ability needed by persons with disabilities (with Down syndrome included) especially because: (i) their workplace activities typically emphasize physical rather than cognitive skills [54]; (ii) muscle weakness can impact their ability to perform everyday activities, including walking, eating, dressing and rising from a chair [17,51]; (iii) life expectancy is increasing for persons with Down syndrome [55] and maintenance of muscle strength is important to lead productive lives [56]; (iv) improving muscle strength may be important in controlling the high tendency for osteoporosis that persons with Down syndrome often demonstrate [50].

Hypotonia and hyper-flexibility, two characteristics of Down syndrome [57,58], have an impact on bone mass, muscular strength and power, gait and motor development [59]. All of these factors lead to the lower strength levels of persons with Down syndrome, but at the same time accent the importance of physical activity. For instance, Daly et al. [60] found that the strength differences between athletes with and without intellectual disabilities are in the range of 4–14% for male and 11–27% for female, being inferior for the athletes with intellectual disabilities. Despite this, much more specific data is needed on high-performance athletes with intellectual disability [23].

Swimming can be one of the activities that can make such a difference. According to Ylmaz et al. [21], aquatic exercises can be a good way of developing physical fitness and motor skill development for children with intellectual disabilities, as aquatics provide a very unique environment for these children. Perán et al. [61] stated that participating in competition is fundamental for individuals with Down syndrome. Although there is research on the effects of aquatic exercises on persons without disabilities, little has been done on persons with intellectual disabilities [62] and more specifically concerning competitive swimming for persons with Down syndrome, so the present study adds new evidence on what physical aspects may distinguish the Down syndrome subjects who are engaged in swimming programs from those who are not.

Comparing the results from the present study with those of Daly et al. [60] with high-performance athletes with intellectual disability, male swimmers with Down syndrome only scored better in the sit-and-reach test (cm) (39.0 ± 8.3 versus 34.0 ± 15.8 finalists and 35.7 ± 7.4 non-finalists) while low scores for the long jump (cm) and sit-ups were obtained (123.3 ± 40.7 for long jump and 17.4 ± 3.8 for sit-ups versus 197.3 ± 26.3 and 23.0 ± 6.8 finalists and 181.9 ± 39.5 and 20.7 ± 4.7 non-finalists). Flexibility was slightly higher for female swimmers with Down syndrome (42.9 ± 4.4 versus 41.0 ± 8.9 finalists and 38.5 ± 8.0 non-finalists) but for the long jump and the sit-ups low scores can be observed (91.0 ± 10.6 and 16.0 ± 2.8 versus 154.6 ± 20.2 and 21.5 ± 6.1 finalists and 157.1 ± 24.9 and 18.2 ± 4.3 non-finalists).

As swimmers with Down syndrome from our study present higher levels of strength than untrained individuals with Down syndrome, we are led to conclude that swimmers have increased muscular hypertrophy, which in turn can reduce hypotonicity and balance dysfunctions and increase bone-mass related parameters [19]. Little is known about the effect of specific strength training in this population. Until recently, swimmers with Down syndrome rarely participated in specific dryland strength training. Van de Vliet et al. [23] studied elite athletes with intellectual disability and pointed out that good levels of fitness seem to be possible for these athletes, and it is likely that the training effect influenced the data. Likewise with athletes, Balic et al. [16] found that the active group of Special Olympians with Down syndrome exhibited significantly higher isometric strength than the sedentary group, also with Down syndrome. They suggested that long term exercise training may enhance physical fitness in individuals with Down syndrome.

Balance in people with Down syndrome is also a component of physical fitness that is usually inferior to the general population or individuals with intellectual disability without Down syndrome [18,63]. Muscle hypotonia may be responsible for balance problems that individuals with Down syndrome usually demonstrate [64]. The delay of maturation of the cerebellum and the relatively small size of cerebellum and brain stem in persons with Down syndrome may also be responsible for the disturbance of balance [65]. Despite these characteristics, individuals with Down syndrome seem capable of improving their balance through physical activity participation, and with this improve their well-being and the quality of life [18]. The swimmers from the current study presented good balance scores and were exceedingly better than the non-swimmers group.

As this was not a training study it is hard to conclude that differences in physical fitness are an outcome of the swimming training. Nevertheless, a study from Querido et al. [66] with six swimmers with Down syndrome evaluated for body composition and physical fitness in 2011 and 2014, found that in 3 years of training, swimmers with Down syndrome improved their physical fitness profile (especially strength) and their body composition characteristics.

In summary, it can be said that: (i) swimmers with Down syndrome present a healthier body composition than untrained individuals with Down syndrome, confirming the first hypothesis; (ii) swimmers with Down syndrome present higher physical fitness values than untrained individuals with Down syndrome, confirming the second hypothesis. This means that swimming educators, parents and/or institutions should see swimming as a sport that can take body composition and physical fitness of Down syndrome subjects to acceptable standards. 

### 4.3. Limitations and Suggestions for the Future

We may point out several limitations to the present study. Larger sample sizes are needed (multi-center), if possible. Information about training characteristics should be more specific (volume, intensity, dry land training), and food intake characterized. Due to the lack of specific equations to estimate fat% and LBM for persons with Down syndrome, general equations were used. Although the participants had previous familiarizations with the tests, future studies should be carried with a test–retest procedure, to ensure that the physical fitness tests are completely understood. In the future it would also be important to perform an intervention program so it would be possible to conclude the effectiveness of swimming training on physical fitness, body composition and other complementary tests.

## Figures and Tables

**Table 1 healthcare-11-00482-t001:** Mean (± SD) and effect size values for the body composition and all variables of the Eurofit Special Test for trained swimmers and untrained subjects.

Variables	Swimmers			Untrained Subjects			Effect Size
	Males (*n* = 14)	Females (*n* = 4)	Total (*n* = 18)	Males (*n* = 9)	Females (*n* = 10)	Total (*n* = 19)	Cohen’s d
BM (kg)	63.8 ± 11.3	53.8 ± 10.4	61.6 ± 11.6	67.5 ± 11.4	62.1 ± 12.8	64.7 ± 12.1	−0.26
Height (cm)	158.4 ± 5.7 a	145.8 ± 6.4	155.6 ± 7.8 c	155.1 ± 5.1 b	141.3 ± 6.7	147.9 ± 9.2	0.91
SS (mm)	47.9 ± 13.4	64.1 ± 22.1	51.5 ± 16.5 c	70.9 ± 23.4	91.8 ± 28.9	81.9 ± 27.9	−1.32
BMI	25.3 ± 3.1	25.9 ± 6.2	25.4 ± 3.8 c	28.3 ± 4.6	32.9 ± 13.3	30.7 ± 10.2	−0.76
Fat%	18.8 ± 3.7 a	29.7 ± 4.7	21.2 ± 6.0 c	24.2 ± 4.2 b	34.7 ± 5.3	29.8 ± 7.1	−1.30
LBM	23.5 ± 3.5 a	17.1 ± 2.6	22.1 ± 4.2	23.1 ± 2.8 b	18.1 ± 2.8	20.5 ± 3.7	0.40
FMI			9.3 ± 3.5 c			13.6 ± 6.1	−0.86
LJ (cm)	123.3 ± 40.7	91.0 ± 10.6	116.1 ± 9.1 c	89.6 ± 45.2 b	56.5 ± 32.6	72.1 ± 41.6	1.10
MB (cm)	384.8 ± 142.3 a	252.0 ± 102.9	355.3 ± 143.4 c	289.5 ± 139.1 b	200.8 ± 74.7	242.8 ± 116.0	0.86
Sit-ups	17.4 ± 3.8	16.0 ± 2.8	17.1 ± 3.6 c	8.7 ± 8.4	7.5 ± 6.7	8.1 ± 7.4	1.56
Speed (s)	5.2 ± 0.5 a	6.4 ± 0.7	5.5 ± 0.7 c	6.7 ± 1.7 b	8.6 ± 1.7	7.7 ± 1.9	−1.53
Flex (cm)	39.0 ± 8.3	42.9 ± 4.4	39.8 ± 7.7 c	29.8 ± 11.7	32.8 ± 10.1	31.4 ± 10.7	0.57
Bal (pts)	5.5 ± 0.7		5.4 ± 0.6 c	4.4 ± 1.1	3.6 ± 1.0	4.0 ± 1.1	1.55

BM = body mass, SS = skinfolds sum, BMI = body mass index, Fat % = percentage of total body fat, LBM = lean body mass, FMI = fat mass index, LJ = long jump, MB = medicine ball, Flex = flexibility, Bal = balance. Differences between genders in swimmers are identified by a, differences between genders in untrained individuals are identified by b and differences between swimmers and untrained individuals are identified by c (*p* ≤ 0.05).

**Table 2 healthcare-11-00482-t002:** Mean (± SD) and effect size percentile values for all variables of the Eurofit Special Test, body mass and height for trained swimmers and untrained subjects.

Variables (Points)	Swimmers (*n* = 18)	Untrained Subjects (*n* = 19)	Mean Difference	Effect Size (d)
Body mass	51.2 ± 21.0	59.8 ± 22.3	−8.7	
Height	28.8 ± 14.5 a	16.0 ± 12.5	12.8	0.95
Long jump	55.9 ± 22.6 a	27.3 ± 27.3	28.6	1.14
Medicine ball	36.7 ± 25.3 a	20.5 ± 22.0	16.2	0.68
Sit-up	83.8 ± 18.8 a	39.6 ± 41.7	44.2	1.37
Speed	83.4 ± 9.0 a	50.2 ± 29.0	33.2	1.81
Flexibility	53.7 ± 13.8 a	33.0 ± 20.2	20.7	1.20
Balance	91.1 ± 9.9 a	53.9 ± 34.1	37.1	1.48

Differences between groups are identified by a (*p* ≤ 0.05).

## Data Availability

Not applicable.

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
