# Peer review of "Swimmers with Down Syndrome Are Healthier and Physically Fit than Their Untrained Peers"

_healthcare, 2023, doi:10.3390/healthcare11040482_

Round 1
Reviewer 1 Report
The paper is very interesting and demonstrates relevant information to the scientific community and health professionals, sports professionals, and family members who have people with Down Syndrome, in order to stimulate the practice of swimming to obtain better health.
I liked the study very much. I have a baby girl with Down Syndrome. I only ask you kindly to change the end of this text (line 153) because you are not verifying by cognitive tests the intellectual disability: "It was found that swimmers with Down syndrome present a healthier body composition and a higher physical fitness score than untrained individuals with the same intellectual disability. Suggestion: "than untrained individuals with Down syndrome" Or "that untrained counterparts" Or "to compare them to untrained peers." or something similar.
Congratulations to the authors!
Author Response
Response to Reviewer 1 Comments
The paper is very interesting and demonstrates relevant information to the scientific community and health professionals, sports professionals, and family members who have people with Down Syndrome, in order to stimulate the practice of swimming to obtain better health.
I liked the study very much. I have a baby girl with Down Syndrome. I only ask you kindly to change the end of this text (line 153) because you are not verifying by cognitive tests the intellectual disability: "It was found that swimmers with Down syndrome present a healthier body composition and a higher physical fitness score than untrained individuals with the same intellectual disability. Suggestion: "than untrained individuals with Down syndrome" Or "that untrained counterparts" Or "to compare them to untrained peers." or something similar.
Congratulations to the authors!
We would like to thank you for the time spent, your valuable comment and effort in giving the clues to improve the manuscript. Your point was considered and changed in the manuscript.

Reviewer 2 Report
Thanks for providing me the opportunity to review this manuscript. See here below some comments:
- The abtsract should contain: introduction, main goal, methodology, results and conclusions
- The introduction is too short regarding swimming as a sport. See here some references that may illustrate you:
Trinidad, A., González‐Garcia, H., & López‐Valenciano, A. (2020). An updated review of the epidemiology of swimming injuries. PM&R, 13(9), 1005–1020. https://doi.org/10.1002/pmrj.12503
Morales, A.T., Tamayo Fajardo, J.A., & González-García, H. (2019). High-Speed Swimsuits and Their Historical Development in Competitive Swimming. Front. Psychol. 10:2639.doi: 10.3389/fpsyg.2019.02639
- Please, divide the method section into: Participants, measures, procedure, data analysis
- Do all the participants participate at the same competition level? Would you mind describing more information about the participants?
- It is needed to describe more in the introduction previous studies similar to the current one as well as the novelty of this study. It is needed to emphasize the novelty.
- It is needed to add practical applications in the discussion.
Author Response
Response to Reviewer 2 Comments
Thanks for providing me the opportunity to review this manuscript.
We would like to thank you for the time spent, your valuable comment and effort in giving the clues to improve the manuscript. We hope that this new version will meet your standards for further publication.
Point 1: The abtsract should contain: introduction, main goal, methodology, results and conclusions
Response 1: We made smooth changes in the abstract to make those sections clear.
Point 2: The introduction is too short regarding swimming as a sport. See here some references that may illustrate you:
Trinidad, A., González‐Garcia, H., & López‐Valenciano, A. (2020). An updated review of the epidemiology of swimming injuries. PM&R, 13(9), 1005–1020. https://doi.org/10.1002/pmrj.12503
Morales, A.T., Tamayo Fajardo, J.A., & González-García, H. (2019). High-Speed Swimsuits and Their Historical Development in Competitive Swimming. Front. Psychol. 10:2639.doi: 10.3389/fpsyg.2019.02639
Response 2: We understand the reviewer point. But none of the requested references where within the main topic of the manuscript (physical fitness in Down syndrome swimmers). Nevertheless you asked in another enquiry to include previous studies similar to this one, we focused mostly on that.
Point 3: Please, divide the method section into: Participants, measures, procedure, data analysis:
Response 3: Changed as suggested
Point 4: Do all the participants participate at the same competition level? Would you mind describing more information about the participants?
Response 4: The trained participants were national and international level. More than half of the participants were part of the National Federation Swimming Team and attended to DSISO International Championships. Based on this, smooth changes were made. We hope that is clear now.
Point 5: It is needed to describe more in the introduction previous studies similar to the current one as well as the novelty of this study. It is needed to emphasize the novelty.
Response 5: Thanks for your suggestion. Few sentences were added for clarity.
Point 6: It is needed to add practical applications in the discussion.
Response 6: It was included as suggested.

Reviewer 3 Report
This manuscript reports on a comparison of swimmers with Down syndrome and untrained patients regarding health status and physical fitness.
It is important that especially in the field of exercise and sport, studies are conducted with special groups, especially persons with disabilities. However, I wonder about the lack of novelty of the present study! The results are internationally known in sports science.
Introduction: The findings that are hypothesized are well known in sports science and medicine. Individuals, whether healthy or disabled, benefit from regular exercise training, which also has a positive effect on the body parameters described by the authors, as well as on physical fitness. Thus, it is trivial that swimmers with down syndrome shows healthier body composition and a higher fitness level than their untrained peers (see also the discussion).
Some older studies are cited that would need to be replaced with newer ones (e.g. l. 40 f; WHO 2022, Chakravarthy et al.)!
Methods: Why was the body mass index chosen? This is not a suitable parameter, especially for persons who are active in sports, as it can remain the same or even increase due to the increase in muscle mass during training. However, this was not the case in this study.
At the beginning of the method description it is unclear which genders were tested. This is only presented in the results (tab. 1). Why was the gender distribution not balanced. The higher proportion of men among the swimmers may be responsible for the better results! Where are the gender effects?
How was it guaranteed that the physical fitness tests were fully understood by the subjects and could therefore be carried out properly?
Author Response
Response to Reviewer 3 Comments
We would like to thank you for the time spent, your valuable comment and effort in giving the clues to improve the manuscript. We hope that this new version will meet your standards for further publication.
Point 1: Introduction: The findings that are hypothesized are well known in sports science and medicine. Individuals, whether healthy or disabled, benefit from regular exercise training, which also has a positive effect on the body parameters described by the authors, as well as on physical fitness. Thus, it is trivial that swimmers with down syndrome shows healthier body composition and a higher fitness level than their untrained peers (see also the discussion).
Response 1: We agree with the Reviewer that these are well known findings in persons without Down syndrome, but studies concerning top level athletes, in this case, swimmers with Down syndrome are very scarce. We made it clearly in the introduction section.
Point 2: Some older studies are cited that would need to be replaced with newer ones (e.g. l. 40 f; WHO 2022, Chakravarthy et al.)!
Response 2: As suggested, WHO was replaced to a recent publication and a more recent study with Down syndrome population was also added.
Point 3: Methods: Why was the body mass index chosen? This is not a suitable parameter, especially for persons who are active in sports, as it can remain the same or even increase due to the increase in muscle mass during training. However, this was not the case in this study.
Response 3: The body mass index alone has some known disadvantages. To diminuish the effect refered by the reviewer the authors measured the four skinfolds and obtained the fat%, lean body mass and fat mass index.
Point 4: At the beginning of the method description it is unclear which genders were tested. This is only presented in the results (tab. 1). Why was the gender distribution not balanced. The higher proportion of men among the swimmers may be responsible for the better results! Where are the gender effects?
Response 4: high level swimmers with Down syndrome in the authors’ country are mainly male. To minimize these differences, the results were converted to percentile scores, facilitating merging of gender groups, as referred in the text.
Point 5: How was it guaranteed that the physical fitness tests were fully understood by the subjects and could therefore be carried out properly?
Response 5: We used the Eurofit special that is a battery of tests already adapted for people with intellectual disability. Some of those tasks were part of their daily routine, but we allowed to get familiar with the battery two weeks before the assessment. For clarity, few word were added accordingly within the methods section.

Round 2
Reviewer 2 Report
.
Author Response
We believe that there are no comments from the Reviewer, nevertheless we would like to thank you once again for your precious contribution on the improvement of this study.